Expression and clinical value of EGFR in human meningiomas

Arnli Magnus B. magnuarn@stud.ntnu.no 1
Backer-Grøndahl Thomas 1
Ytterhus Borgny 1
Granli Unn S. 1 2
Lydersen Stian 3
Gulati Sasha 4
Torp Sverre H. 1 5
1 Department of Laboratory Medicine, Children’s and Women’s Health, Faculty of Medicine and Health Sciences, Norwegian University of Science and Technology (NTNU) , Trondheim , Norway
2 Cellular and Molecular Imaging Core Facility (CMIC), Norwegian University of Science and Technology (NTNU) , Trondheim , Norway
3 Regional Centre for Child and Youth Mental Health and Child Welfare, Department of Mental Health, Faculty of Medicine and Health Sciences, Norwegian University of Science and Technology (NTNU) , Trondheim , Norway
4 Department of Neurosurgery, St. Olavs Hospital , Trondheim , Norway
5 Department of Pathology, St. Olavs Hospital , Trondheim , Norway
Soares Paula
Electronic publication date: 2017 Mar 29
Publication date: 2017
Volume: 5
Electronic Location ID: e3140
Received 2016 Nov 1; Accepted 2017 Mar 2
Copyright: ©2017 Arnli et al.
Copyright year: 2017
Copyright holder: Arnli et al.
License: This is an open access article distributed under the terms of the Creative Commons Attribution License, which permits unrestricted use, distribution, reproduction and adaptation in any medium and for any purpose provided that it is properly attributed. For attribution, the original author(s), title, publication source (PeerJ) and either DOI or URL of the article must be cited.
License URL: https://creativecommons.org/licenses/by/4.0/

Keywords: Brain tumors, Prognosis, TGFα, Survival, Growth factor receptors, EGF, c-erbB1, Immunohistochemistry.

Funding: Norwegian University of Science and Technology (NTNU) The study was funded by the Norwegian University of Science and Technology (NTNU). There was no additional external funding received for this study. The funders had no role in study design, data collection and analysis, decision to publish, or preparation of the manuscript.

==============================
Background

Meningiomas are common intracranial tumors in humans that frequently recur despite having a predominantly benign nature. Even though these tumors have been shown to commonly express EGFR/c-erbB1 (epidermal growth factor receptor), results from previous studies are uncertain regarding the expression of either intracellular or extracellular domains, cellular localization, activation state, relations to malignancy grade, and prognosis.

Aims

This study was designed to investigate the expression of the intracellular and extracellular domains of EGFR and of the activated receptor as well as its ligands EGF and TGFα in a large series of meningiomas with long follow-up data, and investigate if there exists an association between antibody expression and clinical and histological data.

Methods

A series of 186 meningiomas consecutively operated within a 10-year period was included. Tissue microarrays were constructed and immunohistochemically analyzed with antibodies targeting intracellular and extracellular domains of EGFR, phosphorylated receptor, and EGF and TGFα. Expression levels were recorded as a staining index (SI).

Results

Positive immunoreactivity was observed for all antibodies in most cases. There was in general high SIs for the intracellular domain of EGFR, phosphorylated EGFR, EGF, and TGFα but lower for the extracellular domain. Normal meninges were negative for all antibodies. Higher SIs for the phosphorylated EGFR were observed in grade II tumors compared with grade I (p = 0.018). Survival or recurrence was significantly decreased in the time to recurrence analysis (TTR) with high SI-scores of the extracellular domain in a univariable survival analysis (HR 1.152, CI (1.036–1.280, p = 0.009)). This was not significant in a multivariable analysis. Expression of the other antigens did not affect survival.

Conclusion

EGFR is overexpressed and in an activated state in human meningiomas. High levels of ligands also support this growth factor receptor system to be involved in meningioma tumorigenesis. EGFR may be a potential candidate for targeted therapy.

Introduction

Meningiomas account for approximately 30% of intracranial tumors in adults, and despite being predominantly benign, many cases recur (Backer-Grondahl, Moen & Torp, 2012; Perry et al., 2007; Willis et al., 2005). Biomarkers capable of identifying recurrent cases could provide better treatment options and closer follow-up for such patients. A potential candidate is the epidermal growth factor receptor (EGFR), which has been shown to be overexpressed and/or amplified in many human cancers (Libermann et al., 1984; Rokita et al., 2013; Salomon et al., 1995; Torp et al., 1991). Ligands such as EGF (epidermal growth factor) and TGFα (transforming growth factor alpha) bind to the extracellular domain (ECD) of the receptor, leading to effects on differentiation, growth, migration, adhesion, or apoptosis (Yarden & Sliwkowski, 2001).

EGFR and the ligands EGF and TGFα have been shown to be overexpressed in human meningiomas (Andersson et al., 2004; Baxter et al., 2014; Caltabiano et al., 2013; Camby et al., 1997; Carroll et al., 1997; Diedrich et al., 1995; Guillaudeau et al., 2012; Halper et al., 1999; Johnson et al., 1994; Kuratsu et al., 1994; Laurendeau et al., 2009; Lusis, Chicoine & Perry, 2005; Maiuri et al., 2007; Narla et al., 2014; Reubi et al., 1989; Torp et al., 1992; Wernicke et al., 2010), whereas non-neoplastic meningeal tissue show sparse or no reactivity (Carroll et al., 1997; Johnson et al., 1994; Torp et al., 1992), indicating a potential tumorigenic role of EGFR in these tumors. Both membranous and cytoplasmic EGFR immunoreactivity have been described in meningiomas (Guillaudeau et al., 2012; Halper et al., 1999; Horsfall et al., 1989; Johnson et al., 1994; Jones et al., 1990; Smith et al., 2007; Torp et al., 1992), however, the prognostic relevance is scarcely described (Guillaudeau et al., 2012). The receptor also seems to be activated (phosphorylated) (Carroll et al., 1997; Hilton et al., 2016). Moreover, the literature is conflicting regarding expression levels of EGFR across malignancy grades, as some report increasing expression with higher grades (Caltabiano et al., 2013; Diedrich et al., 1995; Halper et al., 1999) and others the opposite or no connection at all (Baxter et al., 2014; Guillaudeau et al., 2012; Jones et al., 1990; Kuratsu et al., 1994; Narla et al., 2014; Wernicke et al., 2010). Furthermore, the prognostic value of expression of either the intracellular or extracellular domain of the receptor is uncertain (Guillaudeau et al., 2012).

The aim of this study was to investigate the expression of the intracellular and extracellular domains of EGFR, as well as its activated status and ligands in a large population-based series of human meningiomas with long follow-up data and the association with histopathological features, meningioma subtypes, malignancy grade, and risk of recurrence.

Methods

Patient selection

The patient selection process has recently been described (Backer-Grondahl, Moen & Torp, 2012). In brief, a search in the electronic patient data files at the pathology department was performed to find all patients consecutively operated for meningiomas between the dates of 01.01.1991 and 31.12.2000 at St. Olavs Hospital—Trondheim University Hospital, Norway. This hospital has the sole responsibility for treating meningioma patients living in Central-Norway. The total population is about 700,000 (2013) (Statistics Norway, 2013). Patients under the age of 18 years or with intraspinal or non-primary meningioma were excluded. In the current study, 10 tumors were not found in the archives, providing a total of 186 primary tumors. Clinical and survival data for patients and histological features have been presented earlier (Backer-Grondahl et al., 2014; Backer-Grondahl, Moen & Torp, 2012). Patients were followed until the time of death or for a maximum of 18 years.

Histological examination

Tumor biopsies were formalin-fixed and embedded in paraffin. Hematoxylin/eosin stained sections were revised microscopically in order to verify diagnoses and to select representative areas for tissue microarray (TMA). Representative areas were defined as meningioma tissue lacking necrosis and with minimal connective- and vascular tissue, hemorrhages, and calcifications. In cases with heterogeneous tissue, areas with different histological patterns were chosen. Microscopic examination was done by two of the authors (MBA and SHT) in collaboration using a Nikon Eclipse 80i microscope.

Tissue microarray construction

Construction of TMAs was achieved with an Alphelys Tissue Arrayer Minicore® 3, AH diagnostics, with corresponding software (TMA Designer2) installed on a dedicated computer. The TMAs were constructed with three cylinders from each biopsy, which has been shown to have the highest level of concordance with whole-tissue sections in other tumors (Fernebro et al., 2002; Hoos et al., 2001; Rubin et al., 2002). The cylinder diameters were 1,000 μm, and spaces between cylinder borders were 600 μm. Each TMA block included a maximum of 24 meningioma cases, yielding 72 cylinders, with three additional cylinders from a liver biopsy for orientation. In total, nine TMA blocks were constructed. 23 tumors were deemed unfit for TMA inclusion due to an insufficient amount of tumor tissue for cylinder extraction. In these cases, whole-tissue sections were cut. The total number of cases analyzed on TMAs was 163.

Immunohistochemistry

Sections were cut with a thickness of 4 μm on a microtome (Leica RM 2255), transferred to glass slides (Superfrost® Plus, Thermo Scientific), dried over night at 37 °C, and later stored in a freezing-unit. Prior to immunohistochemical staining, sections were heated at 60 °C for one hour. After deparaffinization, slides were pre-treated for antigen retrieval with PT Link (Dako) using Target Retrieval Solution, High pH (Dako), except for the EGF antibody where proteinase K was used (10 min). Endogenous peroxidase activity was extinguished using diluted hydrogen peroxide for 10 min. The antibodies used were reactive against the extracellular domain of EGFR (clone EGFR.113, 1:10 dilution, 60 min incubation, mouse monoclonal Ab IgG2a, Novocastra, supplied by Leica Biosystems), the cytoplasmic domain of EGFR (clone EGFR.25, 1:100 dilution, 60 min incubation, mouse monoclonal Ab IgG1, Novocastra, supplied by Leica Biosystems), activated (phosphorylated) EGFR (anti-phospho-EGFR (Tyr1173) antibody, 1:45 dilution, overnight incubation at 4 °C, rabbit monoclonal, EMD Millipore), EGF (clone 10825, 1:180 dilution, overnight incubation at 4 °C, monoclonal mouse IgG1, R&D Systems), and TGFα (anti-TGFα, ab 9585, 1:200 dilution, 60 min incubation, rabbit polyclonal Ab, Abcam). All antibodies were incubated for 30 min with the detection system Dako EnVision™ + HRP, a dextran polymer conjugated with secondary antibodies and horseradish peroxidase. Diaminobenzidin was used as chromogen (2 × 5 min incubation) and hematoxylin as counterstain. Positive controls (placenta, skin, kidney, endometrium or breast) were included in each staining run (Uhlen et al., 2010). In the negative controls the primary antibodies were omitted. Dura and leptomeninges adjacent to the meningiomas served as a reference to normal tissue. EMA (epithelial membrane antigen) (clone E29, 1:500 dilution, 40 min incubation, monoclonal mouse IgG2a, Dako) was used to confirm the presence of meningioma tissue. Sections were stained using a Dako AutostainerPlus.

Immunohistochemical analysis

Immunoreactivity was recorded as a staining index (SI) representing the product of intensity and fraction of positive tumor cells (Torp et al., 2007). Intensity was subjectively evaluated as 0 (no reaction), 1 (weak reaction), 2 (moderate reaction), or 3 (strong reaction). The fraction of positive tumor cells was recorded as 0 (no positivity), 1 (<10% positive cells), 2 (10–50% positive cells), or 3 (>50% positive cells) by a subjective estimate. TMA sections were scanned using an Ariol scanning system (Ariol™ SL-50 3.3) with Genetix analysis system, and analyses were conducted by the investigators on electronic images. An SI-score was calculated for each tumor and used in statistical analyses. TMA tissue cores with <50% remaining tissue after sectioning and staining were excluded from analyses. Whole-tissue sections were evaluated using a conventional microscope (Nikon Eclipse 50i). All cases were analyzed individually by two of the authors, and discrepancies in findings were discussed and a consensus was reached. Each meningioma patient was given an ID unique to this study, and the investigators were consequently unaware of any case-specific clinical data during analysis.

Statistical analysis

The Mann–Whitney U test was used to compare SI between sample groups, according to histological features and malignancy grades. Meningioma subtypes (n = 175, subtypes with 1–2 cases were excluded from analysis) and tumor localization (n = 185, one intraventricular tumor was excluded from analysis) were compared using the Kruskal–Wallis test. When the Kruskal-Wallis test was significant, pairwise comparisons between groups were performed using Dunn’s test and the Hommel adjustment (Dmitrienko & D’Agostino, 2013, page 5191) for multiple comparisons and to preserve the familywise error rate. Cox regression was used in both univariable and multivariable survival analyses based on continuous SI-values. Simpson resection grade (1 and 2 vs. 3 and 4) (Backer-Grondahl et al., 2014), WHO performance status (0 and 1 vs 2, 3, 4 and 5), tumor grade (grade I vs. grade II and III tumors), and age (continuous values) were included as covariates in the multivariable analyses. The proportional hazard assumption was checked by visual inspection of log minus log survival plots. Two measures for survival were investigated: time to recurrence (TTR) and overall survival (OS). TTR was defined as either recurrence or disease-related death (Punt et al., 2007). For TTR, survival was calculated after a maximum of 15 years with follow-up. OS was calculated with a maximum of 18 years of follow-up. Cohen’s quadratic weighted kappa (computed in StatXact 11) was used to assess inter-rater agreement. Hommel adjusted p-values were computed in R. Other analyses were conducted using SPSS version 24.0 (SPSS Inc., Chicago, IL). Two-sided p < 0.05 were considered statistically significant.

Ethics

The study has been approved by the Regional Committees for Medical and Health Research Ethics (REK) (project number 4.2006.947). Waiver of consent was given by REK.

Results

Clinical characteristics are presented in Table S1. Among the 186 meningiomas, 130 (69.9%) were benign, 55 (29.6%) atypical, and one (0.5%) malignant (Table S2). The median age at the date of operation was 59 years (range: 25–86), and the female to male ratio was 2.9:1. Results of immunohistochemical analyses and SI related to histological features are shown in Tables 1 and 2, respectively. Typical immunostainings are shown in Fig. 1. No cases were excluded from the study due to tissue loss or during processing failures. The TMA tissue cores were well preserved after processing, and all three cores were intact for most cases. EMA immunoreactivity was seen in 177/180 meningiomas (98.3%) as well as in normal leptomeninges, which were negative for the other antibodies.

Table 1 Immunohistochemical findings.

Antibody	Measure	All grades (n = 186)	Grade I (n = 130)	Grade II (n = 55)	Grade III (n = 1)	
EGFR25 (ICD)	Percent positive	100	100	100	100	
	Median SI (min–max)	9 (2–9)	9 (2–9)	9 (2–9)	9 (9)	
	Mean SI	7.95	7.94	7.96	9.00	
EGFR113 (ECD)	Percent positive	99.5	99.2	100	100	
	Median SI (min–max)	4 (0–9)	4 (0–9)	6 (2–9)	9 (9)	
	Mean SI	4.89	4.87	4.87	9.00	
Phosphorylated EGFR	Percent positive	99.5	99.2	100	100	
	Median SI (min–max)	9 (0–9)	9 (0–9)	9 (6–9)	9 (9)	
	Mean SI	8.40	8.24	8.78	9.00	
EGF	Percent positive	100	100	100	100	
	Median SI (min–max)	9 (1–9)	9 (2–9)	6 (1–9)	3 (3)	
	Mean SI	7.42	7.62	7.05	3.00	
TGFα	Percent positive	100	100	100	100	
	Median SI (min–max)	9 (6–9)	9 (6–9)	9 (6–9)	9 (9)	
	Mean SI	8.81	8.75	8.95	9.00	
Notes.

Percent positive the percent of cases with a SI of 1 or higher

SI staining index

ICD intracellular domain

ECD extracellular domain

ICD immunoreactivity was found in all meningioma cases at high levels with a median SI of 9.00. Concerning cellular localization, the ICD showed both membranous and cytoplasmic staining, yet with a clear membranous predominance. There was no association between expression of ICD and subtypes (p = 0.765), localization (p = 0.862), and malignancy grade (p = 0.983, Table 2). However, ICD levels were high in tumors lacking psammoma bodies (p = 0.005, Table 2).

Table 2 Comparison of antibody SI and histological features (p-values, 2-tailed exact values from Mann–Whitney U tests).

	EGFR25 (ICD)	EGFR113 (ECD)	Ph-EGFR	EGF	TGFα	
Mitosis 4+ (n = 41)	0.430	0.317	0.057	0.545	0.307	
Brain infiltration present (n = 13)	0.248	0.221	0.574	0.254	1.000	
Sheeting present (n = 14)	0.509	0.035	0.133	0.467	0.604	
Macronucleoli present (n = 11)	0.440	0.684	0.216	0.478	0.622	
Hypercellularity absent (n = 140)	0.200	0.579	0.445	<0.001	0.733	
Small cell change present (n = 18)	0.228	0.259	0.206	0.509	0.380	
Necrosis present (n = 40)	0.416	0.877	1.000	0.210	0.073	
Psammoma bodies absent (n = 61)	0.005	<0.001	0.259	0.168	0.108	
Grade II vs grade I (total: n = 185)	0.983	0.787	0.018a	0.040b	0.113	
Notes.

Values in bold statistically significant

Ph-EGFR phosphorylated EGFR

Brain infiltration n = 65 where brain tissue was observed

a SI significantly higher in Grade II than in Grade I.

b SI significantly higher in Grade I than in Grade II.

p-values between 1% and 5% should be interpreted with caution due to multiple hypotheses. (The present/absent annotations in the first column indicate the characteristic associated with high SI values for bold values only).

Figure 1 Expression patterns for each antibody.

Pictures were taken by MBA using a Nikon eclipse 50i microscope with Nikon DS-Fi2 camera head and Nikon Digital Sight DS-L3 camera controller. Exposure was set to +1.3 and original files were in TIF format. Lettering and merging of pictures were done with Microsoft Paint. (A) EGF, 40×, SI 9. (B) TGFα, 40×, SI 6. (C) EGFR intracellular domain, 40×, SI 9. (D) extracellular domain, 40×, SI 9. (E) EGFR phosphorylated, 40×, SI 9. (F) Non-neoplastic leptomeninges and adjacent tumor tissue (EGFR25), 10x.

Immunoreactivity for ECD was also observed in most cases (99.5%), although the SI was much lower than that observed for ICD with a median SI of 4.00. Membranous immunoreactivity was slightly stronger than cytoplasmic. The ECD was lower in fibrous tumors than in transitional tumors (p = 0.028), atypical (p = 0.044), and meningothelial (p = 0.036). Further, the ECD was higher in tumors presenting sheeting (p = 0.035, Table 2), and in tumors lacking psammoma bodies (p < 0.001, Table 2). There was no association between either malignancy grade or localization and ECD SI (p = 0.787 and p = 0.562, respectively).

Phosphorylated EGFR was found in most cases (99.5%) with a median SI of 9.00. Both membranous and cytoplasmic immunostaining was present in most tumors, although the latter demonstrated higher intensity. There was no association with any histological features. Concerning malignancy, expression was higher in grade II tumors compared to grade I (p = 0.018, Table 2). Further, expression was lower in skull base than convexity (p < 0.001) and falcine tumors (p = 0.038).

Immunoreactivity for EGF and TGFα was strong and seen in all cases with median SIs of 9.00. Statistical analyses concerning histological features and malignancy grades were also performed for grade I and grade II tumors individually (Tables S3 and S4, respectively.)

In univariable Cox regression analyses (Table 3), high ECD expression was significantly associated with reduced survival or recurrence in the time to recurrence (TTR) analysis (HR 1.152, CI (1.036–1.280), p = 0.009). When tumor grades were analyzed individually for this antibody, a similar result was achieved only for benign tumors (n = 38, HR 1.213, CI (1.064–1.383), p = 0.004). In the multivariable survival analysis (Table 4), only Simpson resection grade was significantly associated with worse TTR (HR 2.606, CI (1.558–4.359), p < 0.001), while ECD expression did not show such association (HR 1.099, CI (0.987–1.222), p = 0.084). For ICD, phosphorylated receptor, EGF, and TGFα no associations were found with TTR or OS (Table 3), even when grades were analyzed separately.

Table 3 Survival analyses (Cox regression, univariable).

Antibody	Measure	Time to recurrence (n = 61)a	Overall survival (n = 67)b	
ICD	HR	1.054	1.065	
	CI	0.909–1.222	0.924–1.229	
	p-value	0.484	0.383	
ECD	HR	1.152	1.045	
	CI	1.036–1.280	0.941–1.160	
	p-value	0.009	0.414	
Phosphorylated	HR	1.058	1.120	
	CI	0.862–1.298	0.901–1.392	
	p-value	0.591	0.308	
EGF	HR	0.992	1.038	
	CI	0.884–1.113	0.927–1.163	
	p-value	0.889	0.516	
TGFα	HR	0.989	1.110	
	CI	0.705–1.387	0.754–1.634	
	p-value	0.949	0.596	
Notes.

a The number indicates recurrences or deaths.

b The number indicates deaths.

HR hazard ratio (exp(B))

CI confidence interval (95% CI for exp(B))

Regarding κ-statistics variable scores were established (Table S5).

Discussion

In the present study we have demonstrated that EGFR and its ligands EGF and TGFα are widely expressed in human meningiomas and that the receptor appears to be in an activated state. Increased ECD expression was associated with poorer survival in a univariable analysis. However, it did not prove to be an independent predictor of survival in the multivariable analysis.

Table 4 Multivariable analysis for EGFR ECD (TTR).

TTR, WHO grades I-III.

Variable	HR	95% CI for HR	p-value	
EGFR ECD	1.099	0.987–1.222	0.084	
WHO grade	1.460	0.868–2.455	0.154	
Simpson grade	2.606	1.558–4.359	<0.001	
WHO performance	1.143	0.607–2.151	0.680	
Age	1.015	0.994–1.036	0.161	

Since normal meninges were EGFR immunonegative, our findings support a general upregulation of this receptor in meningioma tissue in accordance with previous results and other studies (Carroll et al., 1997; Di Carlo et al., 1992; Halper et al., 1999; Horsfall et al., 1989; Huisman et al., 1991; Johnson et al., 1994; Kuratsu et al., 1994; Laurendeau et al., 2009; Lusis, Chicoine & Perry, 2005; Maiuri et al., 2007; Narla et al., 2014; Reubi et al., 1989; Torp et al., 1992). These findings are also supported by studies in which EGFR expression in meningiomas was determined by other techniques such as Northern and western blots, polymerase chain reactions (PCR) and ligand-binding studies (Andersson et al., 2004; Carroll et al., 1997; Guillaudeau et al., 2012; Kurihara et al., 1989; Laurendeau et al., 2009; Torp et al., 1992; Weisman, Raguet & Kelly, 1987). As such, immunohistochemistry appears as a reliable and practical method to determine EGFR expression in meningioma tissue. For instance, this may be useful in diagnostics to distinguish between normal and neoplastic meninges as well as to detect meningioma infiltration in soft tissue. The upregulation of EGFR in neoplastic meningeal tissue supports a role of this receptor as a facilitator in the tumorigenesis of these tumors. Whether this is linked to tumor initiation or progression is, however, unclear. The increased levels of EGFR may be a result of enhanced transcription, translation or decreased receptor turnover, but is not likely due to gene amplification, as this is not described in meningiomas (Guillaudeau et al., 2012; Torp et al., 1992).

Few studies have focused on the different expression patterns of the internal and external domains of EGFR in human meningiomas (Guillaudeau et al., 2012). In the present study, nearly all tumors expressed both domains (99.5% for the ECD, 100% for ICD), even though the internal domain had an overall higher intensity. High levels of ICD positivity have also been observed in high-grade astrocytomas and non-small cell lung cancer (Gulati et al., 2010; Mascaux et al., 2011; Torp et al., 2007), although negative immunoreactivity has also been reported (Wickremesekera, Hovens & Kaye, 2010). Concerning the ECD, most studies have shown high expression (Andersson et al., 2004; Baxter et al., 2014; Camby et al., 1997; Diedrich et al., 1995; Guillaudeau et al., 2012), whereas some have recorded lower positivity (Jones et al., 1990). The observed stronger expression of the ICD compared with the ECD may be a result of a fragile ECD damaged during tissue processing or due to various shedding mechanisms (sEGFR) (Perez-Torres et al., 2008). In high-grade astrocytomas such a discrepancy may be due to EGFR gene mutation and truncated receptors, which have not been reported in meningiomas (Aldape et al., 2004; Fan et al., 2013; Guillaudeau et al., 2012; Torp et al., 1992). Contradictory to our findings, Guillaudeau et al. (2012) found higher expression of the extracellular domain compared with the intracellular domain. The mechanisms of these discrepancies remain uncertain, but expression of different EGFR isoforms and various antibodies may be part of the issue (Guillaudeau et al., 2012).

In agreement with others, widespread EGF and TGFα immunoreactivity was observed in all meningiomas (Carroll et al., 1997; Halper et al., 1999; Hsu, Efird & Hedley-Whyte, 1998; Torp, Unsgaard & Dalen, 1993). Both cytoplasmic and membranous localizations were found, with the latter showing lower intensity. However, some did not find any membranous positivity for TGFα (Halper et al., 1999; Hsu, Efird & Hedley-Whyte, 1998). Strong cytoplasmic reactivity may either be due to antibodies binding ligands produced in the tumor cells or internalized ligand–receptor complexes (Tomas, Futter & Eden, 2014). Expression of these EGFR ligands in meningioma tissue has also been demonstrated by other techniques such as Northern and protein blots, PCR, in situ hybridization, and radioimmunoassays (Carroll et al., 1997; Halper et al., 1999; Torp, Unsgaard & Dalen, 1993), supporting the applicability of immunohistochemistry to investigate the presence of these ligands as well. Furthermore, these findings support the existence of autocrine/paracrine growth loops in human meningiomas.

EGFR expression alone does not reveal the activation status, so clarifying whether the receptor is activated or not is relevant. There are sparse data on this issue in meningioma tissue (Carroll et al., 1997; Hilton et al., 2016), and according to our findings the activated receptor is expressed at high levels in most tumors. In their study, Hilton et al. found significantly higher expression of phosphorylated EGFR in tumor tissues compared with non-neoplastic tissues, and high expression of downstream signaling molecules (Hilton et al., 2016). The literature is conflicting concerning EGFR and malignancy, as some have found that high expression is positively related to tumor grade (Caltabiano et al., 2013; Diedrich et al., 1995; Halper et al., 1999) whereas others have found the opposite or no differences (Baxter et al., 2014; Guillaudeau et al., 2012; Jones et al., 1990; Kuratsu et al., 1994; Narla et al., 2014; Wernicke et al., 2010). However, these previous studies have not investigated the activated receptor. Thus, our results suggest that the expression level of the activated receptor can be useful in meningioma grading.

Both membranous and cytoplasmic EGFR immunoreactivity were observed in the meningioma tissue, however, membranous reactivity dominated in most cases for both the ECD and ICD. Distinguishing between these patterns of immunostaining was often problematic, mostly due to high cellularity and fibrous growth. For this reason it seems inappropriate to make such a distinction in human meningiomas, although it may be of clinical relevance in other human malignancies such as renal cell carcinoma (Pu et al., 2009). A study on colorectal cancer, however, demonstrated that cellular EGFR localization was unrelated to clinicopathological parameters or patient outcome (McKay et al., 2002).

Despite the uniform histological appearance of meningiomas, we observed both heterogeneous immunoreactivity in the form of hotspots and homogeneous EGFR immunoreactivity, pointing to meningiomas as heterogeneous in this regard. EGFR expression also varied with regard to meningioma subtypes, suggesting that EGFR may play different roles in tumorigenesis of these variants. Concerning histology, the association between high expression of the internal and external domains and absence of psammoma bodies is interesting as the latter favors poorer survival (Backer-Grondahl et al., 2014). Further, imaging studies suggest that calcified meningiomas have a slower growth rate compared with uncalcified tumors (Nakamura et al., 2003; Nakasu et al., 2005). Accordingly, if there exists a connection between microscopic- (psammoma bodies) and macroscopic calcification, EGFR immunostaining may act as a marker for fast growing tumors.

Concerning survival, only high expression of ECD was significantly associated with decreased TTR, also when benign meningiomas were evaluated separately. Thus, high expression of ECD may be an indicator of recurrence-prone benign meningiomas. This is in contrast to other studies, which report better survival with high levels of ECD and similarily, poorer survival with low expression of ECD (Guillaudeau et al., 2012; Smith et al., 2007). Others report no association between EGFR expression and survival (Caltabiano et al., 2013). Contrary to our findings, Maiuri et al. (2007) found higher EGFR positivity in non-recurrent benign meningiomas. Further, we observed no association between antibody expression and OS. These discrepancies may be due to different antibodies, tumor size, observation time and endpoints. The expression of the ECD, however, was not significant in the multivariable analysis. ECD SI is therefore not an independent predictor of survival for meningioma patients. Thus, the prognostic value of EGFR in human meningiomas appears uncertain and needs to be further clarified.

The key strengths of this study are a population-based study design with long observation time and a large number of patients. Limitations are its retrospective profile and the subjective nature of immunohistochemistry. To compensate for interobserver variability, all cases were evaluated by two observers independently. In addition, the weighted kappa-statistics showed moderate or substantial agreement (Landis & Koch, 1977), suggesting reproducible results. Another uncertainty is the antigenicity of the archived tumor samples (Economou et al., 2014), which may vary between tumors since the inclusion period was over 10 years. The use of TMAs has potential advantages and disadvantages, and our evaluation of the immunostainings might have been interfered by heterogeneous expression patterns. To compensate for this we tried to include TMA cores representing the varying histological patterns. All in all, TMAs have benefits in many aspects such as reproducibility and uniform assay conditions (Camp, Neumeister & Rimm, 2008; Voduc, Kenney & Nielsen, 2008).

Conclusions

EGFR and its ligands are frequently overexpressed in human meningiomas compared with normal meninges, and the receptor is generally in an activated state. These findings support the role of this growth factor system in the tumorigenesis of these tumors. Accordingly, EGFR is a potential candidate for targeted therapy. Regarding clinical value, only overexpression of the external domain of EGFR was associated with prognosis in univariable analyses. It was not, however, significant in the multivariable analysis, and as such EGFR status appears to have minor significance as a prognosticator.

Supplemental Information

Data S1 EGFR and meningiomas - raw data

List of all tumors with relevant data used in this study.

Click here for additional data file.

Table S1 Clinical data

Summary of key clinical records and includes tumor grade, localization, resection grade, patient age and gender.

Click here for additional data file.

Table S2 Grade and subtype distribution

The number and percentage of tumors across malignancy grades and their subtypes.

Click here for additional data file.

Table S3 Comparison of antibody SI and histological features for grade 1 tumors

The table shows p-values (2-tailed exact values) from Mann-Whitney U tests when comparing staining index in tumors with certain histological features to tumors lacking these features. Only grade I tumors are included in these tests.

Click here for additional data file.

Table S4 Comparison of antibody SI and histological features for grade 2 tumors

The p-values (2-tailed exact values) from Mann-Whitney U tests when comparing staining index in tumors with certain histological features to tumors lacking these features. Only grade II tumors are included in these tests.

Click here for additional data file.

Table S5 Kappa statistics

The weighted kappa statistics between observers for each antibody used in the study.

Click here for additional data file.

The authors would like to thank prof. Anna Mary Bofin MD, PhD for her assistance and advice in the study, Øyvind Salvesen PhD for statistical assistance, and prof. Ivar Skjåk Nordrum MD, PhD and Rosilin Varughese MD, PhD for constructive criticism on the manuscript. A special thanks to Kathrin Torseth PhD, Camilla Bjørk Setsaas, and Ingunn Nervik at the Cellular and Molecular Imaging Core Facility (CMIC), Department of Laboratory Medicine, Children’s and Women’s Health at the Norwegian University of Science and Technology (NTNU) where the procedures for this article were performed.

List of abbreviations

EGFR epidermal growth factor receptor

ECD extracellular domain

ICD intracellular domain

HER human epidermal growth factor receptor

EGF epidermal growth factor

TGFα transforming growth factor alpha

TMA tissue microarray

EMA epithelial membrane antigen

SI staining index

TTR time to recurrence

OS overall survival

HR hazard ratio

PCR polymerase chain reaction

Additional Information and Declarations

Competing Interests

Author Contributions

Human Ethics

Data Availability

The authors declare there are no competing interests.

Magnus B. Arnli performed the experiments, analyzed the data, wrote the paper, prepared figures and/or tables, reviewed drafts of the paper.

Thomas Backer-Grøndahl contributed reagents/materials/analysis tools, reviewed drafts of the paper.

Borgny Ytterhus and Unn S. Granli performed the experiments, contributed reagents/materials/analysis tools, reviewed drafts of the paper.

Stian Lydersen analyzed the data, contributed reagents/materials/analysis tools, reviewed drafts of the paper.

Sasha Gulati wrote the paper, reviewed drafts of the paper.

Sverre H. Torp conceived and designed the experiments, analyzed the data, wrote the paper, reviewed drafts of the paper.

The following information was supplied relating to ethical approvals (i.e., approving body and any reference numbers):

The study has been approved by the Regional Committees for Medical and Health Research Ethics (REK) (project number 4.2006.947). Waiver of consent was given by REK.

The following information was supplied regarding data availability:

The raw data has been supplied as a Supplementary File.

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
