# Peer review of "Expression and clinical value of EGFR in human meningiomas"

_PeerJ, doi:10.7717/peerj.3140_

## Round 0.1 · original submission · Major Revisions

· Academic Editor

Major Revisions

Please address all the comments raised by the reviewers (including in the attached PDF). Please remake and improve Figure 1.

Reviewer 1 ·

Basic reporting

Introduction and background should be improved. It lacks to show the context of the work. It is not clear, after reading the introduction, what the authors will do to increase the general pool of information. It should be stated in this section what is currently unknown and which gaps in the literature the authors want to fill and their meaningful. For example, the relevance of studying expression patterns of internal and external EGFR domains. Moreover, it should be written in a way that engage reading and not with telegraphic sentences. For example, the idea of the sentence: “Moreover, the literature is conflicting regarding malignancy” is not perceptible.

The references are missing an important recent work, that should be included in the introduction and discussion: Activation of multiple growth factor signalling pathways is frequent in meningiomas. Neuropathology. 2016 Jun;36(3):250-61.

The authors should also improve Figure 1. The authors state in the "Methods" section that "Dura and leptomeninges adjacent to meningioma served as reference to normal tissues", a picture showing both negativity in these areas and positivity of tumor tissue would improve the information retrieved from the figure. Also, the figure would benefit with pictures with higher magnification, eg. In the form of inserts. Actual magnification is small and makes it hard to see clear membrane or cytoplasmic staining. Also the letters of the pictures should be made bigger. The authors should also add the magnification to the legend of the figure or a scale bar to the pictures. Figure legend is also missing staining index (SI) level of each figure.

The authors should also describe in the text the meaningful of table 6.

Experimental design

As stated before, the rationale and benefit of this work to literature should be clearly stated on the introduction, instead of limited to abstract and discussion.
The investigation seem to have been conducted rigorously and to a high technical standard. Methods are globally well described, however manufacturer should be indicated in all reagents.

Validity of the findings

The generated data is interesting, confirms previously published works and adds to the global knowledge of EGFR expression and activation in meningioma.

Additional comments

No additional comments

Reviewer 2 ·

Basic reporting

The paper by Arnli et al describes the expression of EGFR-related molecules in a series of meningiomas. This is a well conducted study, with a large sample size and accurate immunohistochemistry analysis, which is important to understand the role of EGFR in human meningiomas. The paper is generally well written, although there are minor typos that I took the liberty of correcting in the PDF document (marked in green).

Experimental design

The study contains original research and the problem is clearly stated. The methodology is adequate, described in detail, and the procedures were rigorously conducted. The study is in conformity with ethical standards.

Validity of the findings

The results and robust, controlled and in line with the question. Some clarifications are needed in parts of the text (see comments to the author).

Additional comments

The most striking observation is the positivity of the 5 antibodies in virtually all cases that were analysed. Although this clearly shows that EGFR plays an important role in the pathogenesis of meningiomas, it has the disadvantage of not being very informative. I think that the paper would benefit from having a more detailed analysis on the topography of the stainings, namely whether there are cases that have remarkable membrane or cytoplasmic expression and whether there is any particular subgrouping, such as cases that show a particularly strong expression of the molecules.

I have highlighted in yellow some parts of the text, together with an explanatory comment, directly in the PDF document. In addition to those, my general comments are:

1. The number of Tables may be shortened and be provided as supplementary data, in particular Tables 1, 2 and 6.
2. Table 4 shows the analysis of the associations between antibody expression and histological features in the entire series, but it would be important to have the same analysis within Grade I and Grade II tumours, separately. If such analysis is little informative, then it would not require a new table, but still should be mentioned in the text.
3. There is some inconsistency between what is written in the Results section and what is included in the Tables.
4. Authors should be more clear about the meaning of the TTR analysis, namely on the association between high ECD expression and reduced survival (see comments in the text).

Annotated reviews are not available for download in order to protect the identity of reviewers who chose to remain anonymous.

---

## Round 0.2 · Minor Revisions

· Academic Editor

Minor Revisions

Please address the alteration suggested by Reviewer 2 concerning multivariate analysis.

Reviewer 1 ·

Basic reporting

no comment

Experimental design

no comment

Validity of the findings

no comment

Additional comments

The authors have addressed all the relevant issues in the revised version of the manuscript.

Reviewer 2 ·

Basic reporting

No additional comments to those in the first review

Experimental design

No additional comments to those in the first review

Validity of the findings

No additional comments to those in the first review

Additional comments

- Page 12, line 288, the sentence “In the multivariable survival analysis (Table 4), only Simpson resection grade was significant (HR 2.606, CI (1.558 to 4.359), p<0.001) and not for the ECD (HR 1.099, CI (0.987 to 1.222), p=0.084)“ should be made clearer. For example: “In the multivariable survival analysis (Table 4), only Simpson resection grade was significantly associated with worse OS (HR 2.606, CI (1.558 to 4.359), p<0.001), while ECD expression did not show such association (HR 1.099, CI (0.987 to 1.222), p=0.084).”

---

## Round 0.3 · accepted · Accept

· Academic Editor

Accept

Thank you for publishing in PeerJ